# Increasing the density of passive photonic-integrated circuits via nanophotonic cloaking

Bing Shen[1], Randy Polson[2] & Rajesh Menon[1]

Photonic-integrated devices need to be adequately spaced apart to prevent signal cross-talk. This fundamentally limits their packing density. Here we report the use of nanophotonic cloaking to render neighbouring devices invisible to one another, which allows them to be placed closer together than is otherwise feasible. Specifically, we experimentally demonstrated waveguides that are spaced by a distance of $\sim \lambda_0/2$ and designed waveguides with centre-to-centre spacing as small as 600 nm ($< \lambda_0/2.5$). Our experiments show a transmission efficiency $> -2$ dB and an extinction ratio $> 15$ dB over a bandwidth larger than 60 nm. This performance can be improved with better design algorithms and industry-standard lithography. The nanophotonic cloak relies on multiple guided-mode resonances, which render such devices very robust to fabrication errors. Our devices are broadly complimentary-metal-oxide-semiconductor compatible, have a minimum pitch of 200 nm and can be fabricated with a single lithography step. The nanophotonic cloaks can be generally applied to all passive integrated photonics.

[1] Department of Electrical and Computer Engineering, University of Utah, 50 Central Campus Drive, Salt Lake City, Utah 84112, USA. [2] Utah Nanofabrication Facility, University of Utah, Salt Lake City, Utah 84112, USA. Correspondence and requests for materials should be addressed to R.M. (email: rmenon@eng.utah.edu).

Planar lightwave circuits (PLC) have significant advantages over electronic circuits such as large bandwidth[1,2], absence of the Joule effect[1,2] and higher immunity to interference, among many others. However, the main disadvantage of PLC is their considerably lower density compared with integrated electronics. There are several options to increase the integration density of PLC. One can shrink the footprint of the component devices. Various methods have been proposed to decrease device dimensions including the application of plasmonics[3–5] or of nanophotonics[6–9]. We have previously demonstrated an integrated nanophotonic polarization beamsplitter with a footprint $2.4 \times 2.4\,\mu m^2$, which is at least an order of magnitude smaller than comparable integrated devices that have been demonstrated experimentally before[7]. A second option to increase integration density is to combine the function of multiple devices into a single compact device. Examples of such multi-functional devices include polarization-splitting grating couplers[10], mode-converting polarization splitters[7] and a transformation-optics-based beam shifter[11]. A third option for enhancing integration density is to decrease the spacing between the individual devices. Waveguiding of light in the plane of the PLC is one of the most fundamental functions. However, the integration density of waveguiding is limited by the leakage of light from one waveguide to its neighbour (cross-talk), if the spacing between them is too small. Song et al.[12] proposed a method to decrease this spacing without considerably increasing cross-talk. However, a general method to decrease the spacing between various devices has not been demonstrated. On the other hand, cloaking to prevent detection has been proposed using numerous technologies[13–15]. Zografopoulo and Prokopidis[16] proposed a method for integrated cloaking based on plasmonics, which exhibits considerable parasitic absorption losses due to metal. Integrated all-dielectric cloaks employing conformal mapping were experimentally demonstrated before[17,18]. However, these cloaks typically exhibit footprints of hundreds of micrometers.

Here we apply cloaking to shield the closely spaced devices so as to enable them to be integrated at a much higher density that is otherwise feasible. An inverse-design algorithm is employed to design the integrated cloak[7,19–21] with a footprint of just a few micrometres. We fabricated and characterized cloaked waveguide pairs that exemplify our general approach. Furthermore, our approach is generally applicable to various integrated photonic components.

## Results

**Design.** The concept of 'digital metamaterials' that was previously introduced was applied to design the integrated cloaking devices. The design algorithm is detailed in our previous publications[22–25]. In general, we discretize the device area, say $7 \times 0.5\,\mu m$, into hundreds of pixels, each pixel of size $100 \times 100\,nm$. There are two possible states for each pixel: silicon denoted as '1' or air where silicon is etched away and denoted as '0.' As a result, our device can be exclusively represented by a binary sequence. By toggling the subwavelength pixels between the two states using an iterative optimization technique, we are able to design photonic devices with useful functions. The 100 nm feature size can be readily achieved with advanced photolithography used in the semiconductor industry and our devices are complimentary-metal-oxide-semiconductor (CMOS) compatible. Here we apply this design technique to two different device scenarios. First, we design a nanophotonic cladding cloak that prevents the cross-talk between two closely spaced single-mode waveguides. We experimentally demonstrate that waveguides with a centre-to-centre spacing as small as $0.8\,\mu m$ ($\lambda_0/1.94$) are feasible. This would effectively double the integration density of PLC, as the conventional minimum centre-to-centre spacing between parallel waveguides is $\sim 1.5\,\mu m$[26]. In the second scenario, we designed a nanophotonic cloak that prevents cross-talk between a single-mode waveguide and a closely spaced micro-ring resonator. This device configuration is commonly used for filters and such cloaks can be quite useful for integrating multiple filters into a small area.

**Experiments.** The nanophotonic cloaks for closely spaced waveguides along with the reference waveguides are illustrated in Fig. 1. The cross-section of each waveguide is $0.3 \times 0.3\,\mu m$, the centre-to-centre spacing between the two parallel waveguides is $0.8\,\mu m$ and the design wavelength $\lambda_0 = 1,550\,nm$. The signal is launched in the bottom waveguide from the left propagating to the right. To prevent the signal from leaking to the neighbouring (top) waveguide, we designed a nanophotonic cloak in the cladding region (between the two waveguides). The cloak is confined in an area of $0.5 \times 7\,\mu m$. The minimum feature size of the cloak is $100 \times 100\,nm$. The devices for the transverse electric (TE) and transverse-magnetic (TM) polarizations are illustrated in Fig. 1a,c, respectively. The corresponding steady-state intensity

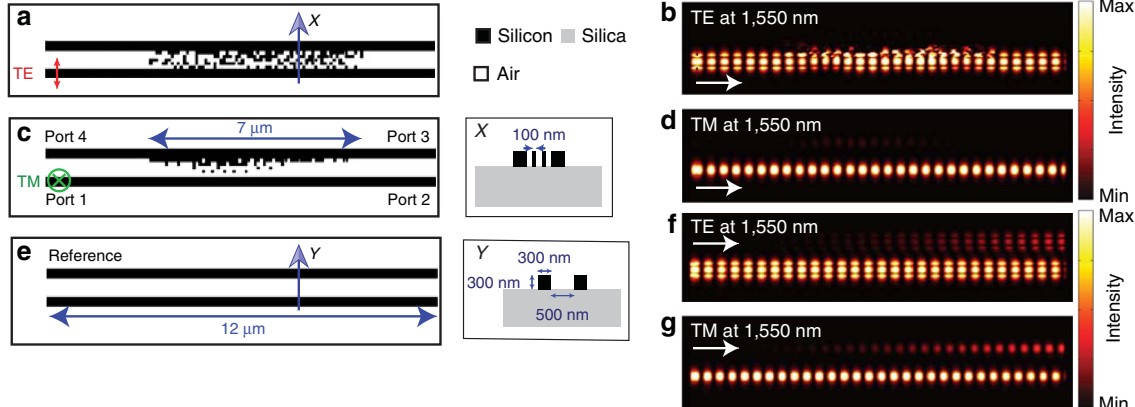

**Figure 1 | Nanophotonic cloaks for closely spaced waveguides.** The geometry and the simulated steady-state intensity distribution at $\lambda_0 = 1,550\,nm$ are shown for TE (in-plane) (**a,b**) and for TM (out-of-plane) (**c,d**) polarizations, respectively. The inset $X$ shows the cross-section of the waveguide and the nanopillars. The corresponding images for the reference device (without the nanophotonic cloaks) are shown in **e–g**. In each case, light is launched in the bottom waveguide propagating from left to right. The white arrows in each figure indicate the light propagation direction. The temporal evolutions of these fields are included in the Supplementary Movie 1 (cloaks) and Supplementary Movie 2 (reference devices).

distributions in the waveguides are illustrated in Fig. 1b,d, respectively. In both cases, it is clear that there is no light leakage from the bottom waveguide to the top, even though the centre-to-centre spacing between them is almost $\lambda_0/2$. In other words, the nanophotonic cloak essentially renders the bottom waveguide invisible to the top waveguide.

For reference, we also simulated the device when no nanophotonic cloak is present as illustrated in Fig. 1e. The same reference geometry was used for both polarizations. The reference steady-state intensity distributions for TE and TM polarizations are illustrated in Fig. 1f,g, respectively. As expected, a large fraction of the light launched in the bottom waveguide (from left to right) is coupled into the top waveguide.

Devices were fabricated using a combination of optical lithography and focused-ion-beam lithography as described previously[7]. The fabrication procedure is also detailed in the Supplementary Methods. Optical patterning via the Heidelberg

µPG 101 is used to generate the pattern for the large structures including the input/output multimode waveguides interfacing the lensed fibre, multi-mode to single-mode tapers and so on. The dual-beam focused-ion beam lithography tool, FEI Helios 650, is used for fabricating the fine features. Scanning-electron micrographs of the fabricated devices are shown in Fig. 2a,b for TE and TM polarizations, respectively.

The measurement setup used to characterize the devices is similar to that described in our previous study[7] and is included in the Supplementary Fig. 1. The detailed characterization process is discussed in the Supplementary Methods. In general, light from an infrared laser goes through a polarization controller (PC) before being coupled to the waveguide via a lensed fibre. The PC in the input path is used to rotate the input polarization state and we use an on-chip polarizer to confirm the polarization state of the light coupled to the waveguide[7]. After transmission through the device, the light is collected by another lensed fibre and goes

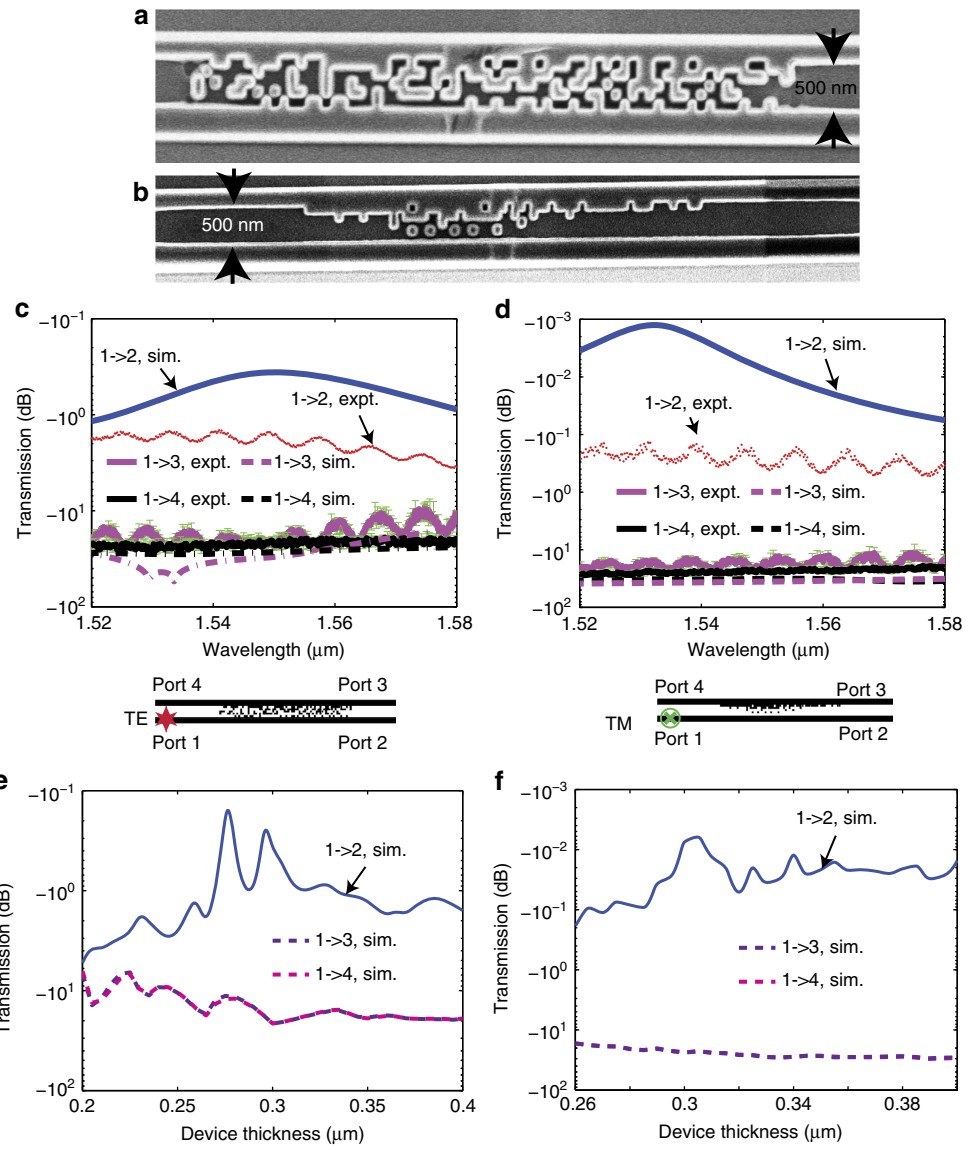

**Figure 2 | Experimental demonstration of cloaking in closely spaced waveguides.** Scanning-electron micrographs of cloaked-waveguide pairs for (**a**) TE and (**b**) TM polarizations. Simulated and measured transmission efficiencies for (**c**) TE and (**d**) TM cloak. Simulated transmission efficiencies for various device thicknesses for (**e**) TE and (**f**) TM polarizations. The naming of each port and source position is illustrated in the insets in the bottom of **c,d**. The green error bars in **c,d** illustrate the fluctuation of measurement data, measured as the differences between the maximum and minimum cross-talk for each individual wavelength.

through another PC and polarizer before being absorbed by the photodetector. The PC in the output path is used to align the polarization plane of the output light with the polarizer and the polarizer is used to select the polarization component of the output light to be measured. Normalizing the transmission of the cloaked waveguides to those of the reference waveguides provides the transmission efficiency, which is plotted as a function of wavelength in Fig. 2c,d for the TE and TM polarizations, respectively. The averaged transmission of two reference waveguides fabricated on the same wafer as the cloaked devices is used for normalization. It is noteworthy that only one cloaked device for each polarization was fabricated and measured. Owing to the constraints in our fabrication process, variations in the devices (such as sidewall roughness) can affect the performance. Nevertheless, this demonstration serves as an effective proof-of-principle of the cloaking technique. In each figure, we have plotted the transmission efficiency, which is defined as the efficiency with which a signal launched at port 1 (left, bottom waveguide) reaches its intended destination port 2 (right, bottom waveguide) as a function of wavelength. Dashed red lines denote the measured spectra, whereas solid blue lines show the corresponding simulations. We have also plotted the cross-talk spectrum, defined as the fraction of light launched at port 1 that ends up at the unintended destinations port 3 (right, top waveguide) and port 4 (left, top waveguide). The green error bars in each figure illustrate the fluctuation of measurement data primarily due to low signal-to-noise ratio. Sidewall roughness in the waveguides also introduces a certain level of uncertainty in our measurements, which will be mitigated when these devices are fabricated using commercial lithographic processes.

The measured TE transmission efficiency at the design wavelength (1,550 nm) is $-1.519$ dB, whereas the corresponding simulated value is $-0.362$ dB. The simulated cross-talk at design wavelength (1,550 nm) is $-28.5$ dB averaged over both directions ($1\rightarrow4$ and $1\rightarrow3$), whereas measurement confirms an average cross-talk of $-21.1$ dB. The measured average TM transmission efficiency at the design wavelength (1,550 nm) is $-0.336$ dB, compared with the simulated value of $-0.0071$ dB. The measured average TM cross-talk at $\lambda = 1,550$ nm is $-22.9$ dB, whereas the simulated value is $-35.87$ dB.

The nanophotonic cloak was designed to render the bottom waveguide invisible to the top waveguide. However, we noticed that for the TM device the same cloak is able to render the top waveguide invisible to the bottom waveguide. In other words, if the signal is launched from port 4 (left, top waveguide), we confirmed that the vast majority of the signal ends up at its intended destination port 3. Details of the simulations are included in the Supplementary Fig. 2. For TE cloak, the simulated transmission efficiency is $> -1.169$ dB within the bandwidth of 60 nm. The transmission efficiency for TM cloak remains unchanged within the band of interest, which is approaching 100%. Such a large operating bandwidth is possible, because a number of guided-mode resonances (rather than a single resonance) are responsible for the cloaking effect.

Numerically evaluated transmission efficiencies for TE and TM cloak under various device thicknesses (Fig. 2e,f) demonstrate their strong robustness to fabrication errors. Specifically, the TE cloak can tolerate variation in device thickness of 83 nm ($-32$ to $+51$ nm), if the transmission efficiency is allowed to fall 1 dB from the value at design thickness (0.3 μm). The corresponding device thickness range for the TM cloak is larger than 140 nm ($-40$ nm to over $+100$ nm). Performance for the TM cloak with device thickness below 0.26 μm is not evaluated, as TM mode is not supported in such waveguides. We also performed numerical analysis of scaling errors in the dimensions of the device

(shown in the Supplementary Fig. 3) and conclude that fabrication precision of $\sim12$ nm is sufficient to maintain transmission efficiency $>90\%$. Such lithographic precision is far easier to achieve than what is required in commercial integrated electronics.

**Symmetric cloaks for waveguides.** As a next step, we designed cloaks that render both waveguides invisible to one another, the so-called symmetric case. The results are summarized in Fig. 3. The average TE transmission efficiency (port 1 to port 2 or port 4 to port 3) is over $-0.969$ dB over a bandwidth $>30$ nm as indicated in Fig. 3j. The corresponding TM transmission efficiency is $> -0.458$ dB over the entire bandwidth (150 nm) as indicated in Fig. 3k. In both cases, the cross-talk (port 1 to port 3, port 1 to port 4, port 4 to port 1 or port 4 to port 2) is $< -12.8$ dB over the entire bandwidth from 1.5 to 1.65 μm and the cross-talk at the design wavelength is below $-22$ dB.

**Increasing the waveguide propagation length.** Although the cloak is designed for a finite length of waveguide, we could extend the waveguides to any length simply by repeating the cloaks. To illustrate this principle, the nanophotonic cloak from Fig. 3d is repeated three times with a gap of 3.3 μm, which gives a total length of 32.6 μm as shown in Fig. 3g. The corresponding steady-state intensity distributions are shown in Fig. 3h,i for TM polarization with signal launched in the top (port 4) and bottom (port 1) waveguides, respectively. The signal energy is confined in the corresponding waveguide without being coupled to the neighbour even after a propagation length of 32.6 μm. The simulated transmission efficiency at 1,550 nm is $-0.872$ dB and $-0.101$ dB for signal launched in the top (port 4 to port 3) and bottom (port 1 to port 2) waveguides, respectively. In both cases, the cross-talk at 1,550 nm is $< -16$ dB. Such propagation length and extinction ratio are sufficiently large to warrant its practical applications in PLCs. As discussed in the Supplementary Fig. 4, the smallest spacing between waveguides that we were able to achieve so far is 0.3 μm for TE polarization with centre-to-centre spacing of 0.6 μm (Supplementary Note 1).

**Improved optimization algorithm.** As mentioned above, propagation length of tens of micrometres is demonstrated without significant energy loss. Such cloak designs can find many useful applications where short cloaking distance is needed, for example, in connections between neighbouring integrated photonic devices or two arms of a Mach–Zehnder interferometer. We can further improve the performance by upgrading the design algorithm to a particle-swarm-based technique as described in the Supplementary Methods[27]. As a result, we were able to design devices with transmission efficiency as high as $-0.1739$ and $-0.0017$ dB for TE and TM polarizations, respectively. The $-0.0017$ dB transmission corresponds to a propagation loss as low as 1.41 dB cm$^{-1}$. These results are summarized in Fig. 4, where the centre-to-centre spacing is 0.8 μm, the same as our previous devices. The simulated cross-talk is $-33.8$ and $-36.9$ dB for TE and TM polarizations, respectively.

Further analysis reveals that the transmission efficiency is almost independent of the length of the cloaking regions. In principle, we could design millimetre-long cloaking region with negligible insertion loss as long as the correspondingly large computation capability is available.

**Cloaking ridge waveguides.** For the cloak designs mentioned above, fully etched waveguides were used. This is due to the fact that the cloak patterns can be fabricated at the same time as the

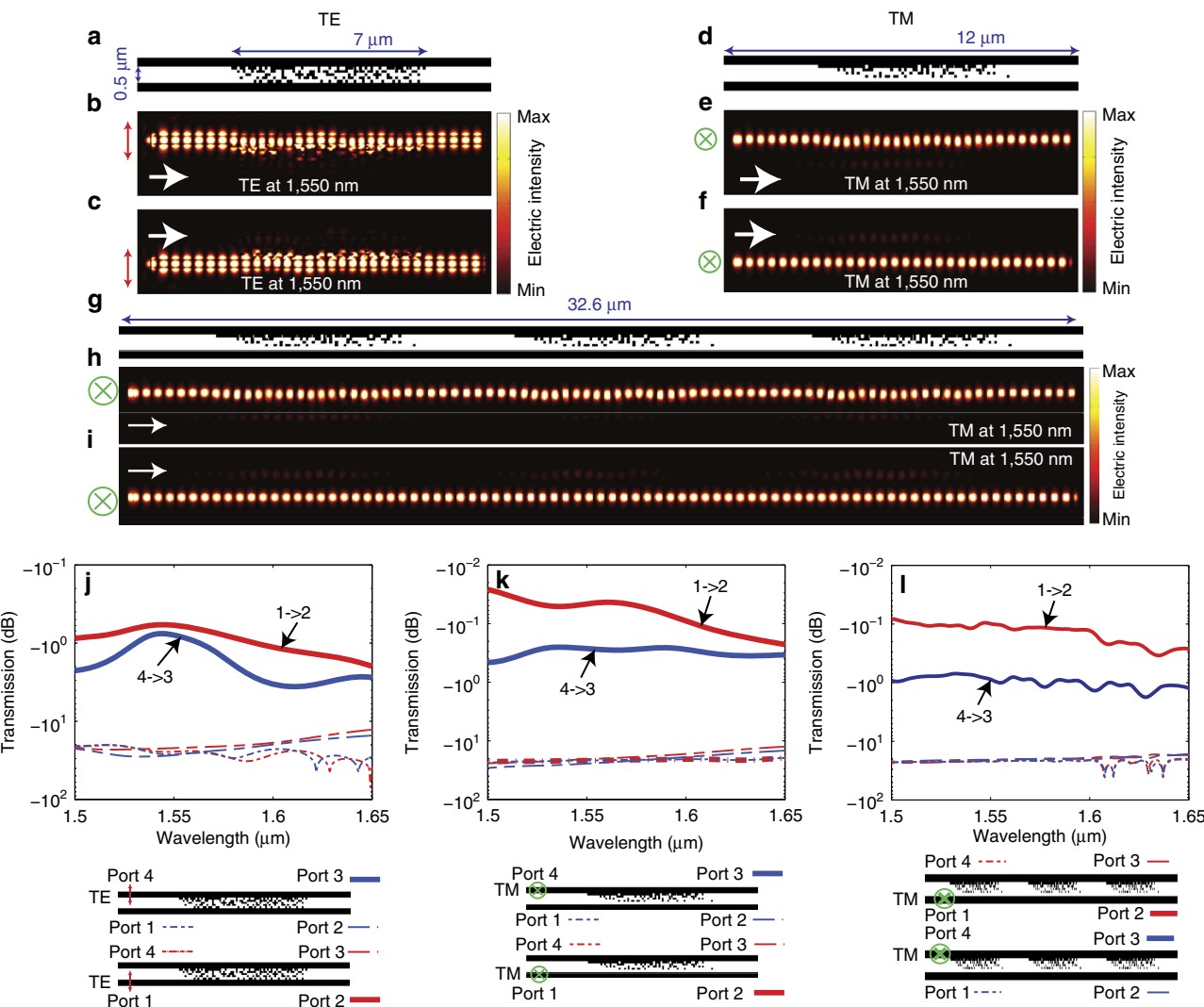

**Figure 3 | Symmetric nanophotonic cloaks for dense waveguides.** The geometries of the devices are shown in **a,d** for TE and for TM polarizations, respectively. The simulated steady-state intensity distributions at $\lambda_0 = 1,550$ nm are shown for (**b**) TE input at port 4, (**c**) TE input at port 1, (**e**) TM input at port 4 and (**f**) TM input at port 1. The simulated efficiencies are shown in **j,k**, for TE and TM polarizations, respectively. (**g**) The geometry of the cloak for waveguides with three repeated units. The simulated steady-state intensity distribution at $\lambda_0 = 1,550$ nm for such repeated cloaks are shown for (**h**) TM input at port 4 and (**i**) TM input at port 1. The simulated efficiencies are shown in **l**. In each case, light is propagating from left to right. The temporal evolutions of these fields are included in Supplementary Movie 3 (TE cloak), Supplementary Movie 4 (TM cloak) and Supplementary Movie 5 (TM cloak with three repeated unit cells).

waveguides, necessitating only a single lithography step. Furthermore, these are easier to fabricate, as chemical dry etching can stop at the oxide layer once the top silicon layer is etched away. However, totally etching away the top silicon layer may be a problem when extending our method to active devices, where a thin slab layer underneath the silicon nanowire is needed to form a PN junction. To demonstrate the versatility of our method, cloak designs based on a ridge waveguide with a slab layer underneath are demonstrated next. The thicknesses of the ridge and the slab layers are 250 and 50 nm, respectively[28]. The centre-to-centre spacing of the waveguides is 0.8 µm. The designs and performance are summarized in Fig. 5. Transmission efficiencies of −0.201 and −0.087 dB at design wavelength (1,550 nm) are achieved for TE and TM polarizations, respectively. The corresponding cross-talk values are well below −23 dB for both cases. In summary, the performance is comparable to that of the fully etched waveguide-based cloaks.

**Cloaking micro-ring resonators.** In addition to waveguides, the nanophotonic cloaking principle can be readily extended to resonators, to enable their very-large-scale integration. To illustrate the generality of our method, we designed a nanophotonic cloak that can render a waveguide invisible to a neighbouring micro-ring resonator. Micro-ring resonators are commonly used in integrated channel-drop or channel-add filters[26,29,30]. In most applications, light is coupled into the resonator via a waveguide that is placed in close vicinity to the ring. However, if another waveguide is placed close to the micro-ring, the two optical components would work as a coupled system with functionality that is different from that of either one working independently, which is illustrated in Fig. 6a,b.

We designed a nanophotonic cloak that allows a waveguide to be placed at a gap of only 300 nm from the micro-ring and essentially renders the waveguide invisible to the micro-ring. The geometry of the device is illustrated in Fig. 6c. The footprint of the cloak is $0.3 \times 6$ µm. The steady-state intensity distribution for

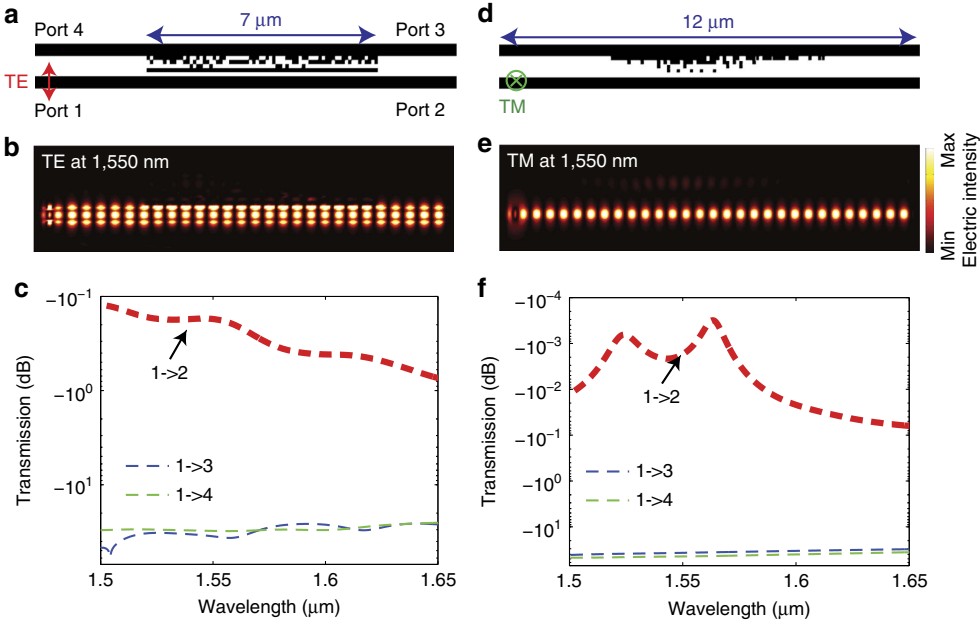

**Figure 4 | Cloak designs with upgraded algorithm.** The geometry, the simulated steady-state intensity distribution at $\lambda_0 = 1{,}550$ nm and wavelength-dependent transmission efficiencies are shown for TE (in-plane) (**a–c**) and for TM (out-of-plane) (**d–f**) polarizations, respectively. The temporal evolutions of these fields are included in the Supplementary Movie 6.

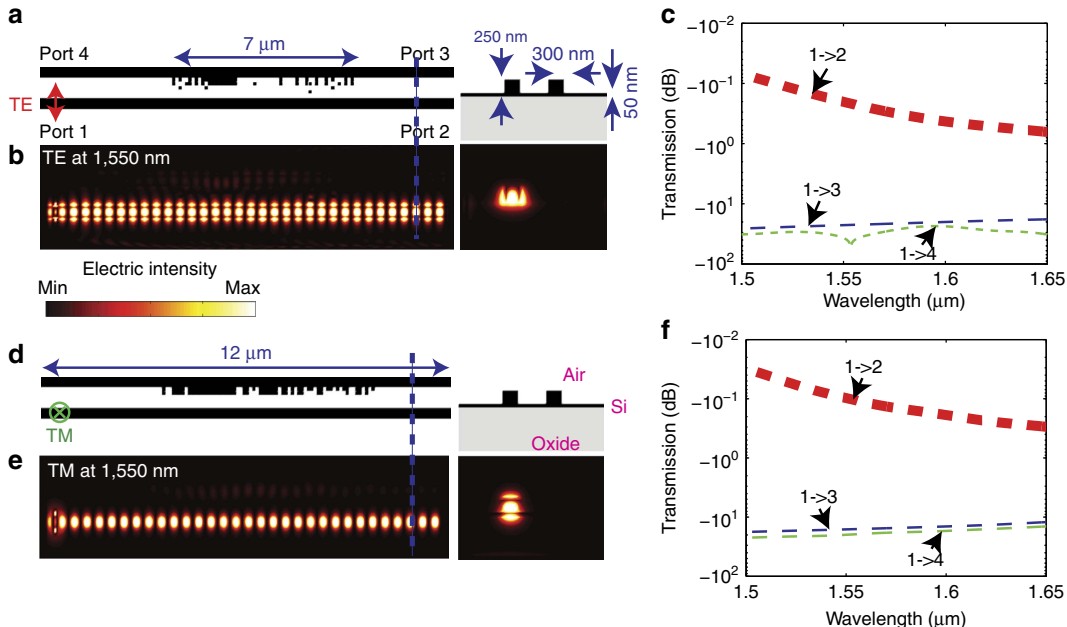

**Figure 5 | Cloak designs for ridge waveguides with a bottom slab layer.** The geometries, the simulated steady-state intensity distributions at $\lambda_0 = 1{,}550$ nm and wavelength-dependent transmission efficiencies are shown for TE (in-plane) (**a–c**) and for TM (out-of-plane) (**d–f**) polarizations, respectively.

a signal launched at port 1 is shown in Fig. 6d. Compared with the case without the cloak (Fig. 6b), almost no light is coupled into the micro-ring. The transmission efficiency, which is defined as the fraction of light that reaches port 2, when a signal is launched at port 1 is plotted in Fig. 6e as a function of wavelength (green dot-dashed line). In this case, the cross-talk is the fraction of light that reaches either port 3 or port 4, when the signal is launched at port 1. The simulated cross-talk spectra in both cases are plotted in Fig. 6f. The transmission efficiency at 1,550 nm is 91.5%, whereas the corresponding cross-talk is <0.5%.

Furthermore, the transmission efficiency is over 87.4%, over the entire bandwidth ranging from 1,500 to 1,650 nm.

It is important to verify that the cloak does not interfere with the proper function of the filter for the right waveguide. The blue dashed curve in Fig. 6e represents the fraction of the signal reaching port 3 when the source is at port 4. The red solid curve represents the same efficiency in the case of a micro-ring resonator without a left waveguide. Compared with this reference, the nanophotonic cloak causes a small shift in the resonance frequency ($\sim$2 nm) and a slight change in the extinction ratio

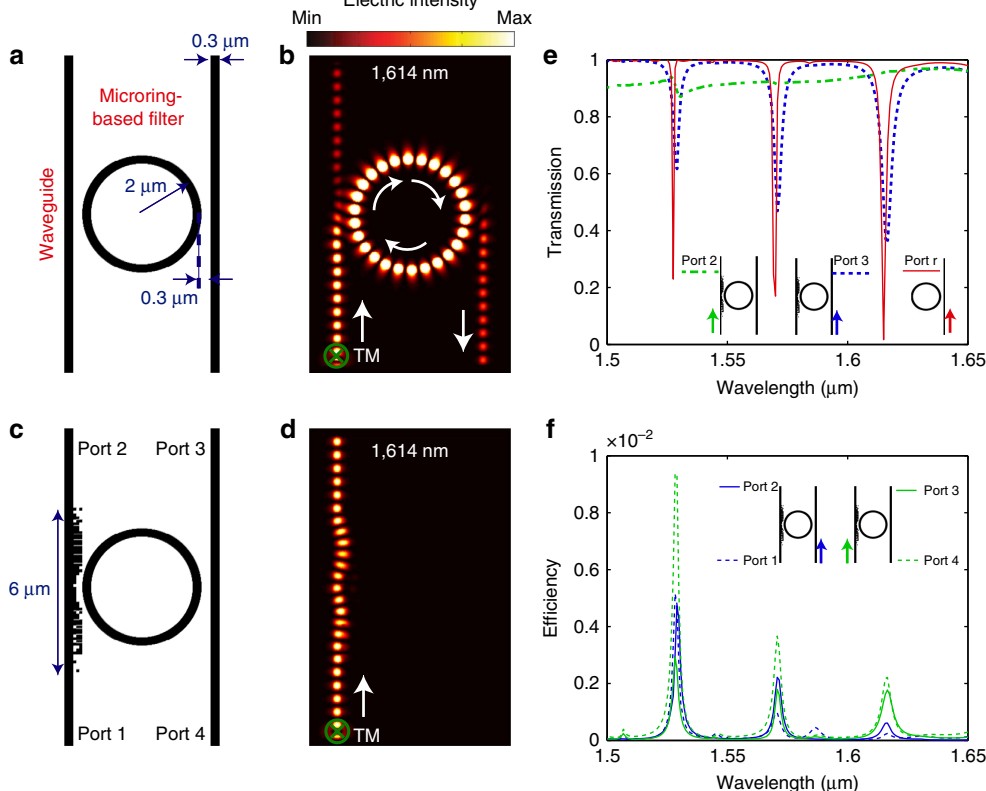

**Figure 6 | Cloak for micro-ring resonator.** (**a**) Geometry of the reference coupled system composed of a waveguide and a micro-ring filter. (**b**) Simulated steady-state intensity distribution for **a** when TM source is launched in the left waveguide. (**c**) Geometry of a system composed of a waveguide, a nanophotonic cloak and a micro-ring. (**d**) Steady-state intensity distribution for the system in **c** when TM source is launched in the left waveguide. (**e**) Transmission spectra of the system in **c** and an individual micro-ring filter as reference. (**f**) Cross-talk for the system in **c**. The temporal evolutions of these fields are included in the Supplementary Movie 7.

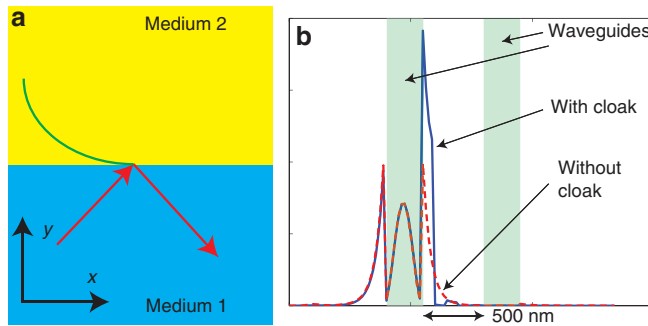

**Figure 7 | Mechanism analysis.** (**a**) Illustration of wave propagating along a waveguide with evanescent wave penetrating from the core (medium 1) to the cladding (medium 2). (**b**) Steady-state intensity distribution along a line through the centre of waveguide perpendicular to the wave propagation direction. Blue solid line represents intensity distribution for cloaked waveguides. Red dashed line represents intensity distribution for a single waveguide as a reference. Green shaded regions indicate the positions of the waveguides.

(∼3 dB lower). The wavelength shift can be compensated thermally[26,30]. Although the cloak was designed to hide the micro-ring from the left waveguide, our simulations indicate that it is fairly effective in reducing any light coupling from the right waveguide to the left waveguide through the micro-ring. Simulation results that verify this behavior are described in the Supplementary Fig. 5.

## Discussion

The mechanism of operation of our devices can be explained as follows. Evanescent waves play a key role in the coupling between neighbouring optical devices. Neighbouring devices can be decoupled and thus invisible to each other by minimizing the evanescent wave's penetration depth into the surrounding medium. For a light wave penetrating into the cladding layer (medium 2) as illustrated in Fig. 7a, the dispersion relation is[31],

$$\frac{\left(k_x^{\parallel}\right)^2}{\varepsilon_y} + \frac{\left(k_y^{\perp}\right)^2}{\varepsilon_x} = (k_0)^2, \tag{1}$$

where $k_x^{\parallel}$ and $k_y^{\perp}$ are the parallel and perpendicular components of the wave vector in medium 2, respectively. $\varepsilon_x$ and $\varepsilon_y$ are the dielectric constants of the medium 2 parallel and perpendicular to the interface, respectively. $k_0$ is the wave vector in freespace. The evanescent wave decay constant in medium 2 is given by

$$k_y^{\perp} = \sqrt{\frac{\varepsilon_x}{\varepsilon_y}} \cdot \sqrt{\varepsilon_x(k_0)^2 - \left(k_x^{\parallel}\right)^2}. \tag{2}$$

The decay rate of the evanescent waves can be enhanced by maximizing the ratio of the dielectric constant parallel to the interface to that perpendicular to the interface in the cladding layer (medium 2). The non-uniform silicon/air pillars are able to create an anisotropic cladding layer as shown in Fig. 1a,c and our nonlinear optimization algorithm aims to maximize this ratio, and thereby minimize the evanescent wave's penetration depth into the cladding layer.

To illustrate the point, the cross-section of the steady-state intensity distribution in Fig. 4b as well as that of a single waveguide without any cloak are shown in Fig. 7b, using blue solid line and red dashed line, respectively. Green shaded regions indicate the positions of waveguides. As can be seen clearly, the introduction of the anisotropic cladding layer enables the evanescent wave outside the waveguide to decay much faster and a much smaller penetration depth into the surrounding medium is observed. The mode length, given by $L_m = \int_{-\infty}^{\infty} W(x)\mathrm{d}x / \max\{W(x)\}$ where $W(x)$ is the energy density of the mode[32], is 8.4 and 3.9 for waveguide with and without cloaks, respectively. This quantitatively demonstrates the anisotropic cladding layer's strong capability in suppressing the penetration of the evanescent components into the cladding layer.

Although our approach to nanophotonic cloaking can increase the integration density of passive photonic devices, we recognize that another important limitation for photonic-integration density is the large size of active devices and hybrid opto-electronics devices. In principle, it is possible to extend our work to these devices in the future and such extension is clearly required for commercial implementation. The figures-of-merit (FOM) for cloaked devices including transmission efficiency and cross-talk will be improved by improving the design algorithm, decreasing the minimum feature size of the nanophotonic cloak and by using industry-standard lithography.

In this study, we designed and experimentally demonstrated the application of nanophotonic cloaking to increase the density of integrated photonics. We were able to place two single-mode waveguides at a distance of almost $\lambda_0/2$ and observe no discernible cross-talk. Furthermore, we were able to design nanophotonic cloaks that allow the placement of a single-mode waveguide next to a micro-ring resonator at a distance $<\lambda_0/2$ with very low cross-talk. These two examples illustrate the generality of our methodology and we emphasize that all passive devices can be cloaked in this manner, enabling a significant increase in the achievable integration density of photonic devices. The cloak design can also be used to reduce the footprint of many individual devices, for example, integrated Mach–Zehnder interferometer with two waveguide arms by reducing the spacing between the arms.

## Methods

**Design algorithm.** The idea of inverse design is already used in nanophotonics[7,19–21]. Our specific design process is as follows. The device is first discretized into square pixels and each of size $100 \times 100$ nm. This initial step ensures that fabrication constraints can be met. For each pixel, there are two possible states: silicon, denoted as '1', and air, where silicon is etched away and denoted as '0'. Next, nonlinear optimization is used to maximize the FOM. For our case, the FOM is the combination of transmission and extinction ratio. Finite-difference time-domain is used to evaluate the transmission and extinction ratio of each design. We used the direct-binary search algorithm and the particle-swarm optimization algorithm for design purposes. All the calculations were carried out on Amazon's web services elastic cluster.

**Device fabrication.** Our devices can be fabricated using a single lithography step. However, our high-resolution lithography tool is limited to patterning small areas. Therefore, we adopted a two-step process. In the first step, photolithography tool Heidelberg μPG101 is used to fabricate the large structures including the input/output multimode waveguides interfacing the lensed fibre. The patterns are transferred into silicon wafer via reactive-ion etching. Next, the smaller features are patterned using focused-ion-beam lithography.

**Data availability.** The data that support the findings of this study are available from the corresponding author upon request.

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

## Acknowledgements

This work made use of University of Utah shared facilities of the Micron Technology Foundation, Inc. Microscopy Suite sponsored by the College of Engineering, Health Sciences Center, Office of the Vice President for Research

and the Utah Science Technology and Research (USTAR) initiative of the State of Utah. This work made use of University of Utah USTAR shared facilities supported, in part, by the MRSEC Program of the NSF under Award Number DMR-1121252. The work is supported by National Aeronautics and Space Administration (NASA) (NNX14AB13G), U.S. Department of Energy (DOE) (EE0005959) and University of Utah.

## Author contributions

B.S. and R.M. conceived and designed the experiments. B.S. and R.P. contributed materials/analysis tools. B.S. performed the experiments. B.S., R.P. and R.M. analysed the data and wrote the paper.

## Additional information

**Competing financial interests:** The authors declare no competing financial interests.

