## [Peer Review File · Nature Communications]

Reviewers' comments:

Reviewer #1 (Remarks to the Author):

The authors present an approach to increase the integration density of silicon based waveguides by placing a metamaterial cloak between the waveguides. The metamaterial is fabricated from silicon and is broadly CMOS compatible.

The approach is moderately novel. However, describing all silicon nanostructures as a metamaterial is tenuous and arguably misleading.

The increase in integration density achieved with the method is unlikely to be significant. Current PIC density is limited by the electronic components rather than the optical. It is likely to be a decade or more before this limit is reached. It should be noted that fully etched structures are not widely used in electro optic silicon devices. Millimetre long Mach-zehnder modulators are today's state of the art. Electro-optic ring resonators are also large -10micron radius. The challenge lies from the need to introduce contacts and doping, which imposes different limits to those of a passive optical device.

The loss is extremely high- 20% loss over a 7um long device. Waveguide losses approaching 1dB/cm are now widely achieved. Even the simulated TM loss is too high. The high losses of the authors' design certainly cancel out any gains due to a higher integration density and probably prevent the operation of a dense circuit. Such isolation may potentially be useful in certain areas of PIC but it is impractical of distances beyond a few 10s microns.

The crosstalk measurement is very important and values of 30dB are probably required for serious applications in optical datacommunications. Lower values are acceptable for an initial demonstration but a route to 30dB needs to be demonstrated.

The claim to CMOS compatibility is an important aspect. In the current version, this is only superficially considered. The authors need to make an analysis of the resolution limits and restrictions of Deep Ultra Violet photolithography and show that their designs fulfill typical design rules, see epixfab.com for example.

Reviewer #2 (Remarks to the Author):

This paper reports the successful demonstration of the cloaking using parallel silicon waveguide. In particular, they demonstrated to avoid the coupling of the two waveguides by adding the digitized sub-wavelength features between the waveguide, regardless of the separation distance to expect significant coupling without the cloaking design. They also demonstrated cloaking of a ring resonator to a silicon waveguide. These demonstrations are quite interesting and will stimulate various readers. The manuscript is well-written and I recommend a publication with optional considerations raised below.

(1) Recommendation to discuss more carefully about the practical applications.

The authors were motivated to increase the density of optical devices by reducing the distance between the waveguides. Indeed, the authors successfully achieved to demonstrate this, however light intensity dropped because of the additional feature. Even if the efficiency was over 80% to across 7um, it will be a significant loss over the propagation of 1cm. Perhaps, I am too practical, but I recommend to explain about the expected propagation loss. Researchers are making significant effort to reduce the propagation loss of Si wire waveguide, and the current state-of-the-art is about 1dB/cm. If the proposed cloaking design will add a significant loss compared with this value, engineers will not employ this cloaking design and just keep the larger distance of the order 2um, which is not a significant penalty in my opinion. In addition, I could not think about a practical situation, where we need to put a silicon waveguide very close to the ring, while avoiding the coupling.

(2) Mechanism;

The authors encouraged to read their previous papers for the mechanism of cloaking, but perhaps, they can outline more about their principle. For me, it looks like the propagating waves are bouncing up and down at the sidewalls of the waveguide. If this internal scatterings are main mechanism, perhaps, we can just expand the width of the waveguide at certain distance and reduce back again, while propagating. This is effectively a coupling to the higher mode, and we do not have to include any sub-wavelength features. I do not know whether this works or not (maybe not), but I hope authors to add more explanation on their tricks, since they are not using plasmonics or other exotic mechanism to break e.g. time-reversal symmetry.

(3) Figures:

It is very difficult to see the differences of lines. For example, in Fig. 2 (c), I can understand the lines for 1->2 ports, but I cannot see other lines. Perhaps, they can prepare insets to explain the color codes.

Having mentioned above criticisms, this work will inspire researchers to lead further innovations. I just wanted to alert not to exaggerate to expect these cloaking designs are universally useful under any situation to increase the density of PIC. Certainly, there will be interesting applications of this technology, and the future promising development is expected.

Response to reviewers with regard to manuscript NCOMMS-16-03915-T.

Response to reviewer 1

We thank the reviewer for very thorough and constructive comments and suggestions. The comments are listed below with our responses.

1. describing all silicon nanostructures as a metamaterial is tenuous and arguably misleading

Response: Although metamaterials are traditionally referred as artificial materials composed of alternating metal and dielectric layer, all-dielectric nanostructures can also be broadly considered as metamaterials in recent literatures as long as these are artificially engineered to have properties that are different than what is seen in the constituent bulk materials. See reference S. Jahani and Z. Jacob, "All-dielectric metamaterials," *Nat. Nanotechnol.* 11(1), 23–36 (2016); Moitra, P. et al. "Realization of an all-dielectric zero-index optical metamaterial," *Nature Photonics* 7, 791–795 (2013). However, the term nanophotonics is more broadly representative of the class of devices that we studies. Therefore, we replaced "metamaterial" with "nanophotonics" in the revised manuscript.

2. The increase in integration density achieved with the method is unlikely to be significant. Current PIC density is limited by the electronic components rather than the optical. It is likely to be a decade or more before this limit is reached. It should be noted that fully etched structures are not widely used in electro optic silicon devices. Millimetre long Mach-zehnder modulators are today's state of the art. Electro-optic ring resonators are also large -10micron radius. The challenge lies from the need to introduce contacts and doping, which imposes different limits to those of a passive optical device.

Response: We agree with the reviewer's comment that PIC is actually a hybrid integration of electric and photonic components for the current stage and PIC integration density is limited by the electronic components due to the need to introduce contacts and doping. However, passive planar lightwave circuits that are exclusively composed of passive optical components can also find many useful applications, e.g. Arrayed Waveguide Grating (AWG) in wavelength division multiplexed (WDM) system. In such circuits, footprint of individual components and spacing between neighboring devices are the key elements limiting the integration density.

In addition, it is true that fully etched structures are not widely used in electro optic silicon devices as mentioned by the reviewer. For our case, devices based on fully etched waveguide is used since the etching would automatically stop at the buried oxide layer after the top silicon layer is totally etched away and the fabrication process is thus simplified when compared to waveguide not fully etched. Most importantly, fully etched devices imply that these devices can be fabricated in **one** lithography step (when the waveguides themselves are patterned). In order to demonstrate that our algorithm can well be applied in active devices, cloaking for waveguide that is not fully etched is demonstrated as well and included in the revised manuscript. The performance is comparable to that of the fully etched waveguide.

3. The loss is extremely high- 20% loss over a 7um long device. Waveguide losses approaching 1dB/cm are now widely achieved. Even the simulated TM loss is too high. The high

losses of the authors' design certainly cancel out any gains due to a higher integration density and probably prevent the operation of a dense circuit. Such isolation may potentially be useful in certain areas of PIC but it is impractical of distances beyond a few 10s microns.

Response: Thanks for the reviewer's comment and a lot of efforts are made to further decrease the propagation loss.

In order to decrease the theoretical loss of the design, we have upgraded our design algorithm. In our previous design algorithm, we used direct binary search (DBS) algorithm to optimize our design. Since DBS is sensitive to the initial guess, the chance of attaining the global optimum is low and it tends to converge to a local optimum. To solve this, we used a particle swarm optimization (PSO) for design. The algorithm is detailed in the revised manuscript. In general, a swarm of particles is initially generated. Each particle updates its position by referring to its own past experience and the social experience of the swarm. During the optimization, a large search space is traversed and the possibility of attaining the global optimum is significantly enhanced. In practice, the local maximum generated from DBS serves as one of the initial particles for PSO in our design and the total iterations are significantly decreased compared to optimization exclusively based on PSO. It shows that the upgraded design algorithm could significantly decrease the propagation loss of the design. The designed insertion loss is as low as -0.0017dB after a propagation length of 12 μ m and equivalent to \sim 1.4dB/cm, which is comparable to the state of the art waveguide propagation loss.

4. The crosstalk measurement is very important and values of 30dB are probably required for serious applications in optical data communications. Lower values are acceptable for an initial demonstration but a route to 30dB needs to be demonstrated.

Response: Thanks for the reviewer's comment. As mentioned in our previous manuscript, the signal to noise ratio is relatively small and we could not precisely measure the extinction ratio. As a result, we did not put the measured extinction ratio in the original manuscript. In the revised manuscript, we used error bars to illustrate the fluctuation of the measurement data, which gives a general idea of the measured extinction ratio. The measured extinction ratio is around 20dB. Further increases in extinction ratio can be achieved with industry standard fabrication techniques compared to the University-level fabrication used in our devices. Even higher extinction ratios can be achieved in devices with larger transmission efficiency, such as the ones designed with our upgraded design algorithm that is included in the revised manuscript.

5. The claim to CMOS compatibility is an important aspect. In the current version, this is only superficially considered. The authors need to make an analysis of the resolution limits and restrictions of Deep Ultra Violet photolithography and show that their designs fulfill typical design rules, see epixfab.com for example.

Response: Thanks for the reviewer's comment. The 100nm feature size (200 nm spacing) of our device can easily be achieved using relatively old 65nm lithography node (193nm Kr-ion optical projection lithography). It is also useful to note that the most advanced lithography node via Extreme Ultra Violet (EUV) can achieve sub-20nm resolution. Therefore, our devices are certainly not limited by industry-standard photolithography. Furthermore, since only one lithography and one step etching process is required, we believe the design generally fulfills the typical design rules. Therefore, we claim our design is generally CMOS compatible.

Response to reviewer 2

We thank the reviewer for very thorough and constructive comments and suggestions. The comments are listed below with our responses.

1. The authors were motivated to increase the density of optical devices by reducing the distance between the waveguides. Indeed, the authors successfully achieved to demonstrate this, however light intensity dropped because of the additional feature. Even if the efficiency was over 80% to across 7 μ m, it will be a significant loss over the propagation of 1cm. Perhaps, I am too practical, but I recommend to explain about the expected propagation loss. Researchers are making significant effort to reduce the propagation loss of Si wire waveguide, and the current state-of-the-art is about 1dB/cm. If the proposed cloaking design will add a significant loss compared with this value, engineers will not employ this cloaking design and just keep the larger distance of the order 2 μ m, which is not a significant penalty in my opinion. In addition, I could not think about a practical situation, where we need to put a silicon waveguide very close to the ring, while avoiding the coupling..

Response: We agree with the reviewer's comment that the loss is significant. In our upgraded design algorithm as shown in the revised manuscript and supplementary information, the theoretical loss is reduced significantly. The designed insertion loss is as low as -0.0017dB after a propagation length of 12 μ m. In addition, we found that designs with a lot of silicon pillars connected with each other is far more robust than that with a lot of individual silicon pillars. We could easily apply the constraint in our design algorithm to enhance the device's robustness to fabrication errors, which could also decrease the experimental propagation loss.

Cloak for microring resonator is shown here because we would like to demonstrate that our cloak design can not only be applied in non-resonant devices (waveguides) but also in resonant devices (resonators). One can imagine geometries of filter banks where multiple ring resonators and waveguides might need to be placed in a small area, where the proximity of unrelated waveguide and micro-ring might be a problem without cloaking. Nevertheless, we only show this as an example, since these two types of devices are the most common components in photonic integration circuits.

2. The authors encouraged to read their previous papers for the mechanism of cloaking, but perhaps, they can outline more about their principle. For me, it looks like the propagating waves are bouncing up and down at the sidewalls of the waveguide. If this internal scatterings are main mechanism, perhaps, we can just expand the width of the waveguide at certain distance and reduce back again, while propagating. This is effectively a coupling to the higher mode, and we do not have to include any sub-wavelength features. I do not know whether this works or not (maybe not), but I hope authors to add more explanation on their tricks, since they are not using plasmonics or other exotic mechanism to break e.g. time-reversal symmetry.

Response: Thanks for the reviewer's comment and the explanation for the cloaking phenomenon is included in the revised manuscript. In general, the cloaking phenomenon is achieved via suppressing the penetration of evanescent wave into the surrounding medium and therefore minimizing the coupling between neighboring waveguides.

3. It is very difficult to see the differences of lines. For example, in Fig. 2 (c), I can understand the lines for 1->2 ports, but I cannot see other lines. Perhaps, they can prepare insets to explain the color codes.

Response: Thanks for the reviewer's comment. We have replot the figures with log scale to make them clear.

Reviewers' comments:

Reviewer #1 (Remarks to the Author):

As the production volumes of photonics is much less than electronics, leveraging the more advanced nodes is non trivial. CMOS compatibility is a much subtler point than most people realise. For example, the full CMOS compatibility of Photonic Crystals has only recently been demonstrated. The authors' structures are a level higher in terms of precision.

The gain in integration density is insufficient motivation. For example silicon AWGs are already very compact, see Selected Topics in Quantum Electronics, IEEE Journal of 12, 1394-1401 (2006). The critical challenge is now phase errors and losses, both of which will be increased by the cloak. Even 1dB/cm additional loss tends to be significant.

In short, the drawbacks and complexity of the designs, more than cancel out the advantages.

Reviewer #2 (Remarks to the Author):

The authors revised their manuscript following the comments from the reviewers. In particular, they have extended their simulations to reduce the propagation loss even further. I do not think they need to emphasize 1.4 dB/cm design in the abstract, but this is just a minor comment. The qualities of the figures are also improved.

In conclusion, I think this is an interesting approach to introduce meta-materials into Si photonics. The authors and reviewers are aware that this technique will not be introduced in a short period of time for practical applications to increase the packing densities of photonic circuits. But, this technique might be used for other applications. It will be useful to publish this paper.

Response to Reviewers' and Editorial comments for Manuscript ID: NCOMMS-16-03915A

Reviewer #1 (Remarks to the Author):

As the production volumes of photonics is much less than electronics, leveraging the more advanced nodes is non trivial. CMOS compatibility is a much subtler point than most people realise. For example, the full CMOS compatibilty of Photonic Crystals has only recently been demonstrated. The authors' structures are a level higher in terms of precision.

Response:

We thank the reviewer for carefully considering our claim of CMOS compatibility. We agree with the reviewer that CMOS compatibility is non-trivial issue from an economic point of view. In conventional electronics chips, the high capital cost of semiconductor manufacturing is amortized over a large volume of chips that are sold. In photonics, such large volumes do not exist, **yet**. However, our claim of CMOS compatibility is purely based upon technical considerations, not economic ones. This claim is supported by the following arguments:

1. Our devices are comprised of only silicon. A passivation layer of silicon oxide or silicon nitride can easily be added on top as needed. So, all the requisite materials are fully compatible with current CMOS manufacturing.
2. The requisite precision of our devices is significantly relaxed from those required for electronics devices. For example, our analysis in Figs. 2 (c) and (d), the device thickness can deviate by many tens of nanometers with only a small degradation of performance. This is consistent with our previous devices, which have demonstrated similar robustness to fabrication errors (see B. Shen, *et al*, *Nat. Photonics*, 9. 378, 2015). Furthermore, we performed a numerical sensitivity analysis on the dimensions of the device in Fig. 4(a) and included this in the revised supplementary information. This analysis clearly shows that fabrication tolerance of 12nm is more than sufficient to maintain the performance of the device within 90% of the designed transmission. From the International Technology Roadmap for Semiconductors (ITRS: <http://www.itrs2.net>), the 3σ precision of the most advanced lithography node is already less than **1nm**. See here for example: http://www.nist.gov/pml/div683/conference/upload/Diebold_final.pdf Therefore, the precision required for our devices can be easily met by already existing commercial processes.
3. The critical dimension (CD) of our devices is 100nm with a minimum pitch of 200nm. This is at least an order of magnitude larger than the CD and minimum pitch of the commercially available lithography processes today, which are <10nm and <20nm, respectively (See Ref: M. Neisser and S. Wurm, ITRS Lithography roadmap: 2015 Challenges, *Adv. Opt. Techn.* 4(4):235-240 (2015)). Therefore, we reassert our claim that our devices are easier to fabricate than electronics devices.

4. It is important to distinguish our devices from Photonic Crystals. In contrast to photonic crystals, our devices rely on numerous coupled guided-mode resonances, which ensure not only broadband performance, but also render our devices to be much more robust to fabrication errors. Photonic crystals and all devices that rely on single resonances (like micro-ring resonators) are extremely sensitive to fabrication errors as the reviewer is most likely very well aware of.

We added additional discussion to illustrate these points in the manuscript.

The gain in integration density is insufficient motivation. For example silicon AWGs are already very compact, see Selected Topics in Quantum Electronics, IEEE Journal of 12, 1394-1401 (2006). The critical challenge is now phase errors and losses, both of which will be increased by the cloak. Even 1dB/cm additional loss tends to be significant.

Response:

We thank the reviewer for pointing us to the reference on AWGs. First, we emphasize that our cloaking technique is independent of the optical functionality. In other words, our methodology can be applied to increase the integration density of **any** passive device. We selected the neighboring waveguides as an example. Second, even though the transverse spacing of AWGs can be small (450nm, still larger than the smallest spacing that we have shown, 300nm), these require separate slab lenses and a very restricted geometry, which render such devices extremely large (for example, 11mm X 16mm in this paper: S. Cheung, *et al*, *IEEE Journal of selected topics in quantum electronics*, 20(4) July/August 2014). In contrast, our cloaked waveguides are completely separate waveguides, whose performance is largely independent of the length of the waveguides. As a result, the overall integration density can be higher by orders of magnitude, and no other separate slab lenses are necessary. Furthermore, our devices are extremely robust to fabrication errors as pointed out above. This is in stark contrast to AWGs, which require extremely precise lithography as pointed out by the reviewer. Finally, we re-iterate that the loss of 1dB/cm is obviously not the limiting factor of our devices. By improving the optimization algorithm and by utilizing smaller pixel sizes (from 100nm to say 25nm), this loss can be dramatically decreased. Our objective with this paper was to point out a novel application of cloaking to increase photonic-integration density. We believe that other scientists will extend our work to come up with devices with much lower losses in the future.

In short, the drawbacks and complexity of the designs, more than cancel out the advantages.

Response:

We disagree with this conclusion of the reviewer. Our goal is to point out a new application of cloaking, whereby one can render one photonic device invisible to a neighbor. As a result, we can increase photonic integration density. We strongly believe that the advantages of this application (increased density, CMOS

compatibility, robustness to fabrication errors, potential for lower losses, etc.) all overshadow any perceived weaknesses. All our designs are based upon rectilinear geometries with critical features of 100nm. These are considerably simpler than the curvilinear geometries usually used in many photonic devices and significantly larger than the geometries in advanced transistors. Therefore, we are a little surprised that the reviewer thinks that our designs are complex!

Reviewer #2 (Remarks to the Author):

The authors revised their manuscript following the comments from the reviewers. In particular, they have extended their simulations to reduce the propagation loss even further. I do not think they need to emphasize 1.4 dB/cm design in the abstract, but this is just a minor comment. The qualities of the figures are also improved.

In conclusion, I think this is an interesting approach to introduce meta-materials into Si photonics. The authors and reviewers are aware that this technique will not be introduced in a short period of time for practical applications to increase the packing densities of photonic circuits. But, this technique might be used for other applications. It will be useful to publish this paper.

Response:

We thank the reviewer for the supportive comments. We will remove the reference to the 1.4dB/cm design in the abstract.

Editorial comments:

- a discussion that will emphasize that this is an approach, which does not have practical implications at this stage within the Si photonics community
- you mention an extinction ratio of 20db please include a pathway for improving on this figure of merit, or indeed any other figures of merit
- Any other discussion relating to realizing the practical implementation of this approach

We included a brief discussion about these points at the end of the manuscript.

- The abstract must be less than 150 words and accessible. It should include the background and context of the work, 'Here we report' or an equivalent phrase, and then the major results and conclusions of the paper.

We revised the abstract based on these guidelines.

- * To comply with our Article templates, the text must be split into:
 - Introduction (<1000 words) which must include the background and rationale

for the work, and the final paragraph should be a brief summary of the major results and conclusions of the paper. The results of the current study should only be discussed in the final brief paragraph of the introduction.

- 'Results' which must be split into sub-headed sections.
- 'Discussion' without subheadings.
- We also require a 'Methods' section, which must be split into sub-headed sections.

* All subheadings in the 'Results' and 'Methods' sections must be <60 characters (including spaces) to comply with our article templates.

We revised the manuscript based on these guidelines.

Other changes:

1. We modified the title slightly to clarify the devices to “Increasing the density of passive photonic-integrated circuits via nanophotonics cloaking.”
2. A new figure S6 was included in the revised supplementary document detailing our analysis of the scaling errors.

All edits are highlighted in the version of the manuscript attached next.